# Imbalance of Peripheral Temperature, Sympathovagal, and Cytokine Profile in Long COVID

**DOI:** 10.3390/biology12050749

**Published:** 2023-05-19

**Authors:** Pablo Fabiano Moura das Neves, Juarez Antônio Simões Quaresma, Maria Alice Freitas Queiroz, Camilla Costa Silva, Enzo Varela Maia, João Sergio de Sousa Oliveira, Carla Manuela Almeida das Neves, Suellen da Silva Mendonça, Aline Semblano Carreira Falcão, Giovana Salomão Melo, Isabella Boechat Faria Santos, Jorge Rodrigues de Sousa, Eduardo José Melo dos Santos, Pedro Fernando da Costa Vasconcelos, Antonio Carlos Rosário Vallinoto, Luiz Fábio Magno Falcão

**Affiliations:** 1Center for Biological Health Sciences, State University of Pará, Belém 66087-670, Brazil; 2Tropical Medicine Center, Federal University of Pará, Belém 66055-240, Brazil; 3School of Medicine, São Paulo University, São Paulo 01246-903, Brazil; 4Laboratory of Virology, Institute of Biological Sciences, Federal University of Pará, Belém 66075-110, Brazil; 5Institute of Health Sciences, Federal University of Pará, Belém 66075-110, Brazil; 6Instituto Evandro Chagas, Secretaria de Vigilância em Saúde, Ministério da Saúde, Ananindeua 67030-000, Brazil; 7Laboratory of Genetic of Complex Discasse, Institute of Biological Sciences, Federal University of Pará, Belém 66075-110, Brazil; 8Graduate Program in Biology of Infectious and Parasitic Agents, Federal University of Pará, Belém 66075-110, Brazil; 9Departamento de Arbovirologia e Febres Hemorrágicas, Instituto Evandro Chagas, Ananindeua 67030-000, Brazil

**Keywords:** long COVID, thermography, autonomic nervous system, cytokines

## Abstract

**Simple Summary:**

In this study, we looked at how persistent inflammation affects peripheral body temperature and sympathovagal balance in individuals with long COVID. Increased temperature and reduced heart rate variability were directly related to the increase in inflammatory cytokines and reduction in anti-inflammatory cytokines. We identified a possible “molecular signature” for long COVID, characterised by a Th17 inflammatory profile with a reduced anti-inflammatory response, resulting in alterations in homeostatic functions and sympathovagal balance.

**Abstract:**

A persistent state of inflammation has been reported during the COVID-19 pandemic. This study aimed to assess short-term heart rate variability (HRV), peripheral body temperature, and serum cytokine levels in patients with long COVID. We evaluated 202 patients with long COVID symptoms categorized them according to the duration of their COVID symptoms (≤120 days, n = 81; >120 days, n = 121), in addition to 95 healthy individuals selected as controls. All HRV variables differed significantly between the control group and patients with long COVID in the ≤120 days group (*p* < 0.05), and participants in the long COVID ≤120 days group had higher temperatures than those in the long COVID >120 days group in all regions analysed (*p* < 0.05). Cytokine analysis showed higher levels of interleukin 17 (IL-17) and interleukin 2 (IL-2), and lower levels of interleukin 4 (IL-4) (*p* < 0.05). Our results suggest a reduction in parasympathetic activation during long COVID and an increase in body temperature due to possible endothelial damage caused by the maintenance of elevated levels of inflammatory mediators. Furthermore, high serum levels of IL-17 and IL-2 and low levels of IL-4 appear to constitute a long-term profile of COVID-19 cytokines, and these markers are potential targets for long COVID-treatment and prevention strategies.

## 1. Introduction

Little is known about the long-term physical effects of severe acute respiratory syndrome coronavirus 2 (SARS-CoV-2) infection. Many people present with various symptoms in different organs and systems for long periods after recovering from the coronavirus disease (COVID-19). Known as long COVID-19 syndrome (long COVID) [1,2], the symptoms include dyspnoea and chest pain, headache, anosmia, amnesia, alopecia, anxiety, insomnia, dementia, various muscle symptoms, cardiac arrhythmia [3,4], and autonomic nervous system dysfunction [5,6]. As a result of the ongoing COVID-19 pandemic, the number of people with prolonged COVID-19 symptoms is increasing, as more people contract COVID-19 [7].

It is likely that neuroinvasion by SARS-CoV-2 via peripheral nerve endings and subsequent entry into the central nervous system (CNS) via the retrograde pathway up to the hypothalamus and brainstem structures causes the dysfunction of the autonomic nervous system (ANS) and functions controlled by these structures [8,9,10,11].

The reduction in parasympathetic activation and the consequent sympathetic hyperactivation resulting from autonomic dysregulation may cause damage to the cardiovascular system in patients with long COVID [12], and sustained changes in levels of inflammatory cytokines due to viral infection at the hypothalamic level [13] can result in the thermal dysregulation of the body, and endothelial injury through an inflammatory vasculopathy [14] with possible consequences for the basal metabolism [15]. Understanding these mechanisms can help to identify the organs and systems affected and to predict the level of severity and permanence of certain symptoms that occur over the course of long COVID [16].

These changes can be detected through infrared thermography (IRT), which is used to detect inflammation and perfusion in the human body [17,18], and heart rate variability (HRV), which provides information on the functionality of the ANS [19]. Reduced HRV is a predictor of cerebrovascular and cardiovascular events and a predictor of mortality [20,21]. Thus, this study aimed to assess changes in body temperature, HRV, and changes in levels of pro- and anti-inflammatory cytokines in patients with long COVID for up to 15 months.

## 2. Materials and Methods

### 2.1. Study Design

This is an observational cross-sectional, analytical, controlled, and descriptive study. All participants provided written informed consent. The study was reported according to the Strengthening the Reporting of Observational Studies in Epidemiology (STROBE) guidelines.

### 2.2. Setting, Period of Study, and Population

The study was conducted in patients of both sexes with long COVID admitted to the post-COVID-19 clinical follow-up programme in the Brazilian Eastern Amazon between September 2020 and November 2021. SARS-CoV-2 infection was confirmed according to the criteria established by Raveendran [22], and patients were evaluated at least 30 days after diagnostic confirmation and the onset of COVID-19 symptoms. The exclusion criteria were incomplete data, comorbidities (asthma, chronic obstructive pulmonary disease, heart disease, diabetes mellitus, and cardiac arrhythmia), and the use of antihypertensive drugs, such as beta blockers, that could interfere with HRV. None of the participants had been vaccinated against COVID-19 or reported any SARS-CoV-2 reinfection.

### 2.3. Participants

Non-probabilistic intentional sampling was used to select participants. The sample of 297 participants who had recovered from acute COVID-19 comprised 202 individuals with long COVID symptoms, and 95 individuals without symptoms as controls (Figure 1). Based on the symptom duration, the participants with long COVID were divided into ≤120 days (n = 81) and >120 days (n = 121) groups. To assess the cytokine profiles of participants with long COVID, blood samples were collected from a convenience sample of 155 patients and divided into two groups: long COVID (n = 60) and controls (n = 95). Demographic and clinical data were collected, including sex, age, height, weight, body mass index (BMI) [23], hospitalization, and length of stay.

### 2.4. Heart Rate Variability

HRV and IRT were recorded in the morning with the participant on a stretcher in the supine position, following the criteria described by Ring and Ammer [24]. HRV data were collected over a 10 min period using a Polar™ RS800CX heart rate monitor and a WearLink heart rate sensor (Polar Electro Oy, Kempele, Finland). Data series were collected using Polar ProTrainer 5™ software (Polar Electro Oy, Kempele, Finland), converted to .txt format, and submitted to Kubios HRV 2.2™ software (Kubios Oy, Kuopio, Finland) for HRV analysis. The 5 min stretches that contained greater signal stability over 256 consecutive beats were selected, discarding the initial 30 s and the final 30 s of data collection [25,26,27].

The linear variables analysed for HRV in the time domain were the RR interval (iRR) and the square root of the mean square of the differences between adjacent normal RR intervals (RMSSD), expressed in ms. In the frequency domain, the high-frequency components (HF, 0.15 to 0.40 Hz), low frequency (LF, 0.04 to 0.15 Hz), and the ratio between the low- and high-frequency components (LF/HF) with normal values between 1.0 and 3.0 [28], in normalized units (un) were analysed. In the non-linear analysis, geometric methods were used to present the dispersions of the RR intervals and to obtain the HRV measurements, such as Poincaré standard deviation perpendicular to the line of identity (SD1), Poincaré standard deviation along the line of identity (SD2), SD1/SD2 ratio, and approximate entropy [29,30].

### 2.5. Infrared Thermography

Thermograms were measured using a FLIR ONE Pro™ camera (FLIR Systems, Inc., Wilsonville, OG, USA) with a thermal image resolution of 160 × 120, frame rate of 8.7 Hz, measurement accuracy of ±3 °C or ±5%, field of view of 8 to 14 µm, fixed focus from 15 cm to infinity, minimum focal distance of 0.3 cm, and a thermal sensitivity of 150 mK. The skin emissivity was set at 0.98 for all the acquired measurements with the camera maintained at a distance of 70 cm from the evaluated limb and perpendicular to the acquisition plane.

Before starting data collection, participants remained at rest, sitting for 15 min for acclimatization. Participants were also asked to abstain from coffee, tea, soft drinks, alcoholic beverages, thermogenic and/or stimulants at least 8 h before the analysis, and to avoid using deodorants, antiperspirants, or other cosmetics that could affect the acquired thermographic pattern. Following said acclimatization period, images were taken with participants lying down, barefoot, on a couch in the supine position. Participants were instructed to keep their feet momentarily in slight dorsiflexion and pointing vertically upwards, while a thermal image were taken using a uniform black backdrop on the on the plantar side of the feet, in order to generate a significant difference in contrast between feet and background in the thermal image.

Thermograms were analysed using the FLIR Tools+ software version 6.4 (FLIR System, Stockholm, Sweden). The measurements were made by adding measurement lines that provided the maximum, average and minimum temperatures of each demarcated area. For this research, we used only the average temperatures provided by this software, using the “rainbow” side palette. Five regions of interest (ROIs) were analysed on the dorsal surface. Region D1 corresponds to the distal end of the first toe to the navicular bone; regions D2, D3, and D4 correspond to the distal extremity of the toes up to the metatarsophalangeal joint of the second, third, and fourth toes, respectively; and D5 region corresponds to the distal end of the fifth toe to the cuboid bone (Figure 2).

### 2.6. Serum Cytokine Levels

A 10 mL blood sample was collected by intravenous puncture using a vacuum collection system containing ethylenediaminetetraacetic acid (EDTA) as an anticoagulant. The samples were transported to the Virology Laboratory of the Federal University of Pará, where they were processed to separate plasma and leukocytes. The plasma was used to measure cytokine levels.

Serum cytokine levels were quantified using flow cytometry and Cytometric Bead Array Kit (CBA) Human Th1/Th2/Th17 (BD Biosciences, San Diego, CA, USA) with BD FACS Canto II equipment. All procedures were performed according to the manufacturer’s guidelines. The methodology used is based on beads conjugated with a capture antibody observed according to their respective fluorescence intensities, from the least bright to the brightest (IL-17 < IFN-γ < TNF-α < IL-10 < IL-6 < IL-4 < IL-2).

### 2.7. Statistical Analysis

Categorical variables were reported as counts and proportions, and continuous variables were reported as the mean ± standard deviation (SD) or median with interquartile range (IQR). Normality of the distribution of cytokine levels, peripheral temperature of the feet, and HRV values were analysed using the Shapiro–Wilk test and the homogeneity of variances by Levene’s test. Based on the results of the normality test, the evaluation of variations in the levels of these variables between groups was performed using analysis of variance (ANOVA) with Tukey’s post hoc test (parametric) or the Kruskal–Wallis test with a post hoc Dunn test (nonparametric). Statistical significance was defined as a two-sided *p* < 0.05. Statistical analyses were performed using GraphPad Prism software. RStudio 4.0.2 software was used to assess the correlation (Spearman correlation) between cytokine levels and the post-COVID-19 time (PCT).

## 3. Results

The demographic data and clinical characteristics of study participants are shown in Table 1.

The most prevalent symptoms among groups of long COVID-19 patients are described in Table 2. Symptoms such as amnesia/loss of recent memory/lack of attention, muscle weakness, decreased visual acuity, paresthesia in limbs, dizziness and arrhythmias were more prevalent in patients with up to 120 days of long COVID-19. On the other hand, fatigue, myalgia, anosmia/hyposmia/parosmia, shortness of breath, headache, arthralgia and hair loss were more frequent in patients with more than 120 days of long COVID-19.

Table 3 shows the mean lower limb temperatures and linear and non-linear analyses of HRV in participants with long COVID based on the symptom duration. We found higher mean temperatures in participants in the ≤120 days group than in the other groups, especially in patients in the >120 days group. In addition, the ≤120 days group had reduced global HRV, lower sympathovagal balance, reduced parasympathetic modulation and increased sympathetic modulation by HRV.

An assessment of cytokine levels between the groups showed higher levels of IL-17 (Figure 3A) and IL-2 (Figure 3G) in patients in the ≤120 days; IL-6 (Figure 3E) and IL- 4 (Figure 3F) were higher in the control group than in the other groups. The levels of the cytokines IFN-γ (Figure 3B), TNF-α (Figure 3C), and IL-10 (Figure 3D) did not differ significantly according to the group.

A linear correlation analysis was applied to the variables IL-17, IFN-γ, TNF-α, IL-10, IL-6, IL-4, IL-2, and PCT, in general and stratified by the two groups, ≤120 days (represented by the red colour) and >120 days (represented by the green colour), through the correlation matrix (Figure 4). It presented Spearman’s linear correlation coefficient, the data distribution curve graph, the data dispersion graph, the histogram, the boxplot, and the statistical significance indicated by the presence of an asterisk for each pair of variables. Patients with a PCT ≤120 days showed a positive correlation with IL-6 levels, whereas those with a PCT >120 days showed a negative correlation with IL-17 levels. A correlogram of the total PCT (15 months) revealed a positive correlation for most of the cytokines analysed, in addition to the positive correlation between IL-2 levels and PCT.

## 4. Discussion

Our study used intentional non-probabilistic sampling to select participants and found a reduction in heart rate variability and parasympathetic modulation, especially in participants in the long COVID ≤120 days group, in addition to the increase in peripheral body temperature between 0.1 and 1.6 °C in areas assessed by IRT, in participants in the long COVID ≤120 days group compared to the >120 days group. We also found higher levels of the inflammatory cytokines IL-17 and IL-2 in participants in the long COVID ≤120 days group, and lower levels of the anti-inflammatory cytokine IL-4 in both groups of participants with long COVID compared to the controls.

The majority of participants were women. Middle-aged women have a higher risk of developing persistent symptoms [31], and the difference between the sexes is attributed to immunological and hormonal differences between sexes [32,33], which may explain the higher proportion of women than men in our study. Overweight was also more prevalent in our study participants. In general, obese individuals tend to have a chronic inflammatory profile [34] that can affect endothelial and hormonal functions [35]. Being overweight is one of the risk factors associated with disease severity in COVID-19 as it leads to chronic activation of the immune system, altering immune functions, and host defence mechanisms [23,36,37].

Most studies of long COVID have focus on its symptoms [38,39,40,41,42,43,44], whereas few studies have evaluated the molecular markers of inflammation in patients with long COVID. Sun et al. [45] analysed the plasma levels of cytokines in patients who recovered from the acute phase of COVID-19. Similar to our results, no differences were found in the levels of IFN-γ, IL-10, and TNFα between the groups studied; however, the levels of IL-6 and IL-4 were higher in patients with neurological symptoms than in controls. This relationship was not observed in our study: IL-6 and IL-4 levels were lower in both the long COVID groups than in the controls. The difference between the results of this study and those observed by Sun et al. (2021) may be due to their sample not being well-matched by sex in addition to having a small size. Our study was carried out with a sample six times larger and, therefore, with greater power of analysis.

The persistence of biomarkers [45] and inflammatory cytokines [46,47] in the acute phase of infection can cause an increase in temperature. Our results also showed higher temperatures in patients in the ≤120 days group, which may be directly related to the increase in IL-17 and IL-2 cytokines found in this group of patients. In addition, we found a decrease in temperature in participants in the >120 days group, which was also correlated with a reduction in the levels of these inflammatory cytokines in this group. Forty days post-infection, patients affected with mild or asymptomatic COVID-19 still had significantly elevated levels of several biomarkers involved in inflammation and the stress response [48], further reinforcing the idea of a post-COVID-19 hyperinflammatory state even in mild cases, causing endothelial damage [49,50], which has been reported to persist for up to 3 months in patients with long COVID [51].

Using flow cytometry, it was possible to identify immuno-inflammatory molecular patterns that may impair the thermoregulation and cardiac autonomy by characterizing the cytokine profile, associated with long COVID. We found higher levels of IL-17 and IL-2 and lower levels of IL-4 in participants in the ≤120 days group, suggesting a possible “molecular signature” for increased body temperature and reduced HRV in long COVID characterized by a Th17 inflammatory profile with a reduced anti-inflammatory response mediated by IL-4, that must be confirmed by other studies enrolling a largest sample size. The correlation of cytokine levels in the long COVID group was positive among most cytokines evaluated, and when correlated with PCT, there was a positive correlation between IL-6 and PCT in patients in the ≤120 days group, and a negative correlation between IL-17 and PCT in patients in the >120 days group, suggesting that the inflammatory profile changes over the course of long COVID.

The IRT was widely used in public places during the pandemic despite its controversial effectiveness. In a multicentre study, Briosch et al., 2023 [52], improved a method based on infrared imaging using artificial intelligence for screening febrile and subfebrile people, identifying suspected and confirmed cases of COVID-19 (+) with temperatures below 37.5 °C. Our study is the first to assess long-term COVID-19 using infrared thermography. The use of thermography in the early stages of the long COVID-19 would allow identifying those patients with higher temperatures to monitor their symptoms, allowing the tracking of inflammatory markers and early initiation of a treatment and/or rehabilitation program, avoiding the permanence of this syndrome for many months, as already registered in the literature.

We found higher temperatures in the first and fifth toes than in the second, third, and fourth toes, owing to differences in the arterial supply of the analysed segments, which supports a direct relationship between blood flow and temperature. The dorsal metatarsal artery of the hallux is a continuation of the dorsal foot artery, and that of the fifth toe is a continuation of the lateral tarsal artery. The remaining three toes (second, third, and fourth) on the dorsum of the foot are supplied by the metatarsal arteries, branches of the arcuate artery, which is normally a minor artery formed by a branch of the dorsal and lateral tarsal artery of the foot. Our results are consistent with the results found by Gatt et al. [53], who found higher temperatures in the hallux and fifth toe (30 °C), while the second, third, and fourth toes have a lower temperature.

Our results suggest that the temperature changes found in this study result from the maintenance of high levels of inflammatory mediators causing changes in microcirculation, such as vasodilation, increased vascular permeability, and increased peripheral blood flow [54], and/or result from the neuroinflammation of structures responsible for controlling body temperature such as the hypothalamus, with no relationship with increased sympathetic modulation and sympathovagal imbalance diagnosed by linear and nonlinear variables of HRV, as we will see later.

In individuals with COVID-19, both sympathetic arousal and parasympathetic inhibition may play key roles in increasing the risk of adverse events [55]. Individuals with SARS-CoV-2 infection and comorbidities are at a greater risk of developing severe disease and, consequently have higher morbidity and mortality rates [56,57]. Individuals with comorbidities tend to have an increase in sympathetic discharge and a reduction in parasympathetic tone [58], exerting significant harmful effects on target organs [59], which can worsen when infected with COVID-19 [60,61].

Some studies have reported ANS dysregulation in long COVID [12,33] as a result of increased sympathetic activation and reduced parasympathetic activation. In our study using HRV data, we found significant differences between iRR, SDNN, RMSSD, and HF values in participants in the long COVID ≤120 days group compared to the controls. The balance between sympathetic and parasympathetic activity in the ANS is determined by several structures and activation pathways that, alone or in combination, influence homeostasis. The malfunction of structures such as the preoptic area, hypothalamic paraventricular nucleus, and solitary tract nucleus (NTS), which integrate and communicate the efferent pathways of sympathetic activity to target organs, may play an important role in the impairment of thermoregulation and cardiac autonomy [34,35].

Autoimmune components and persistent inflammation after SARS-CoV-2 infection also appear to affect the ANS [37]. The altered behaviour of the ANS with reduced HRV in patients with long COVID for ≤120 days found in our sample corresponded to the elevated levels of IL-17 and IL-2 cytokines in this group. Pan et al. [28] investigated the trend of HRV changes in the evolution of patients with severe COVID-19, where approximately half of these evolved with an increase in the LF/HF ratio and a reduction in the Standard Deviation of the Normal RR Interval (SDNN) and Standard Deviation of the Five-minute Mean NN Intervals (SDANN) during the post-COVID-19 period, requiring more time to eliminate the virus and recover, indicating a prolongation of the inflammatory state with worse rates of cardiac biomarkers. In the same way as Pan et al. [28], we observed that the RMSSD, SD1, and SD2 values continued to evolve with a reduction in patients in the >120 days group, indicating the maintenance of the parasympathetic imbalance state and possibly of the inflammatory state, which would justify the permanence of this syndrome for periods longer than 6 months in our sample, which could be related to the maintenance of low levels of IL-4 in this group.

The permanence of the state of neuroinflammation, resulting from the neuroinvasive capacity of the virus and the immune response of the host to COVID-19 [62], may be responsible for similar changes in peripheral temperature and cardiac autonomic function during COVID-19, suggesting functional alteration of interconnected structures. In addition to these changes, our study showed CNS tropism through the predominance of neurological symptoms, such as amnesia, muscle weakness, paresthesia, and reduced visual acuity, especially in patients with long COVID for ≤120 days.

Our findings are supported by comparison with age- and sex-matched controls without comorbidities and with no history of SARS-CoV-2 infection, but there are some limitations to our study. First, the IRT was performed only in the extremities of the body, and it is not possible to compare the temperature of the central areas of the patient’s body, which are higher by around 4–5 °C and, therefore, can alter the thermal pattern found in this study. Second, while the thermal camera used in the research can be used to capture thermal images of human skin, it is not designed or validated for medical diagnosis, health assessment, or accurate body temperature measurement. Although this does not diminish the study’s findings or significance. Third, as this is a cross-sectional study, our results do not provide information on the evolution of the disease over time, requiring prospective studies to assess its long-term behaviour.

## 5. Conclusions

Our study found increases in temperature ranging from 0.1 and 1.6 °C in areas assessed by IRT, showed high serum levels of IL-17 and IL-2, low levels of IL-4, and an increase in peripheral temperature and a reduction in HRV in patients with long COVID. The change in peripheral temperature may be related not only to hypothalamic dysfunction, but also to an integral part of a system that controls other homeostatic functions, such as osmoregulation, rhythm generation (e.g., ultradian, circadian), breathing, appetite, sleep, metabolic control, and endothelial damage, as a result of maintaining high levels of inflammatory mediators, altering microcirculation, and increasing vascular permeability. These novel results could contribute to the development of future studies, allowing a better understanding of long COVID. In the future, a comprehensive prospective analysis of the cytokine profile in patients with long COVID and their effects on the ANS and body temperature over time should be conducted with a larger sample size, in order to establish treatment and prevention strategies more suitable for specific groups.

## Figures and Tables

**Figure 1 biology-12-00749-f001:**
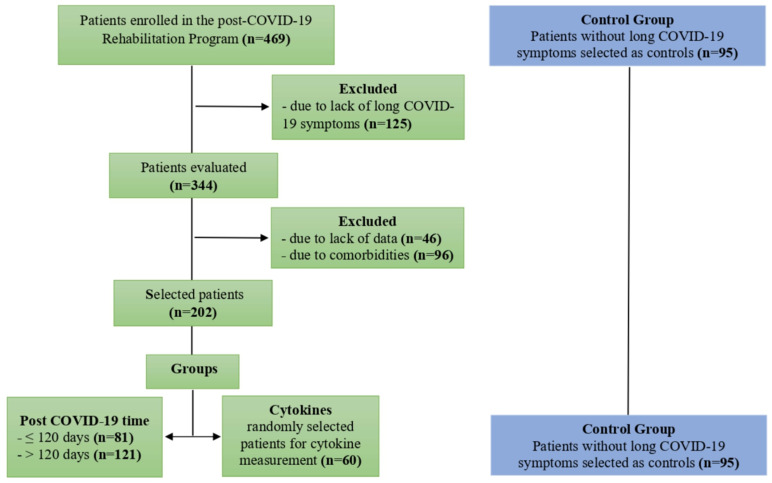
Flow diagram of selection and distribution of study participants.

**Figure 2 biology-12-00749-f002:**
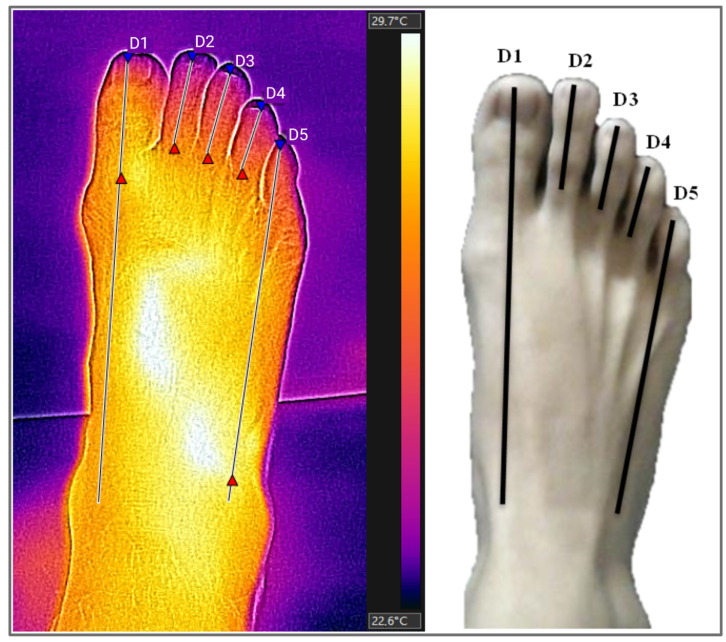
Thermogram and regions of interest (ROIs) on the dorsal surface of the feet. ROIs (regions of interest): D1 (distal end of the first toe to the navicular bone), D2, D3, and D4 (distal extremity of the second, third, and fourth toes up to the metatarsophalangeal joint), D5 (distal end of the fifth toe to cuboid bone). Source: Authors themselves, 2020–2022.

**Figure 3 biology-12-00749-f003:**
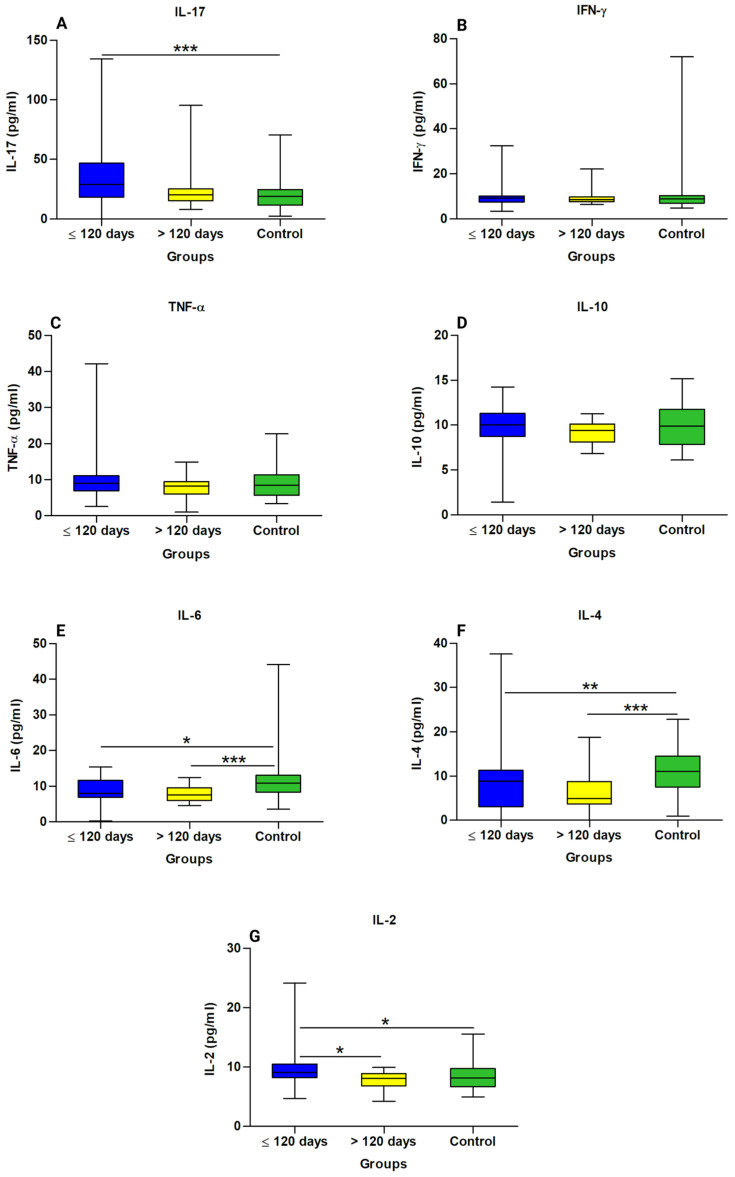
Comparison of cytokine levels in the long COVID subgroups and controls. (**A**) IL-17 Interleukin 17, (**B**) IFN-γ interferon gamma, (**C**) TNF-α tumor necrosis factor alpha, (**D**) IL-10 interleukin 10, (**E**) IL-6 interleukin 6, (**F**) IL-4 interleukin 4, and (**G**) IL-2 interleukin 2. * Statistically significant if *p* < 0.05 (* *p* < 0.05, ** *p* < 0.01 and *** *p* < 0.001), Kruskal–Wallis with Dunn’s post hoc test (nonparametric). pg/mL: Picogram per millilitre.

**Figure 4 biology-12-00749-f004:**
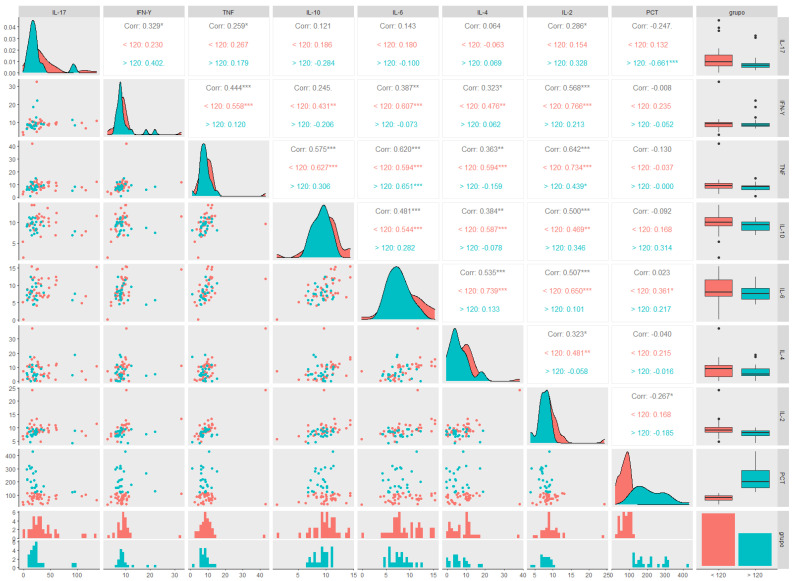
Correlogram between cytokine levels in participants with long COVID and post-COVID-19 time ≤120 and >120 days. (IL-17) Interleukin 17, (IFN-γ) interferon gamma, (TNF-α), tumour necrosis factor alpha, (IL-10) interleukin 10, (IL-6) interleukin 6, (IL-4) interleukin 4, (IL-2) interleukin 2, and post-COVID-19 time (PCT). ≤120 days group (red color) and > 120 days group (green color). * Statistically significant if *p* < 0.05 (* *p* < 0.05, ** *p* < 0.01 and *** *p* < 0.001), Spearman correlation (nonparametric) R = 1: directly proportional correlation, R = 0: no correlation, and R = −1: inversely proportional correlation.

**Table 1 biology-12-00749-t001:** Participants demographic and clinical data.

	N (%)	
Characteristic	Long COVID-19 (n = 202)	Controls (n = 95)
Gender (n, %)		
Men	77 (38.1%)	34 (35.8%)
Women	125 (61.9%)	61 (64.2%)
Age, median (IQR), y	46.5 (38–54)	39 (34–45)
Height median (IQR), m	1.62 (1.56–1.69)	1.63 (1.57–1.69)
Weight median (IQR), kg	75 (66–87)	68 (62.5–81.5)
Classification BMI (n, %)		
Underweight	1 (0.5%)	0
Normal range	48 (23.8%)	29 (30.5%)
Overweight	153 (75.7%)	66 (69.5%)
Long COVID-19 symptomatic period median (IQR), d	114.5 (88.5–243.8)	-
≤120 days	81 (40.1%)	-
>120 days	121 (59.9%)	-
Main symptoms in long COVID-19 (n, %)		
Dyspnea	180 (89.1%)	-
Chest Pain	119 (58.9%)	-
Muscle weakness	149 (73.8%)	-
Tremor	79 (39.1%)	-
Fatigue	171 (84.7%)	-
Myalgia	133 (65.8%)	-
Headache	116 (57.4%)	-
Visual changes	95 (47.0%)	-
Insomnia	109 (54%)	-
Lower limb edema	68 (33.7%)	-
Hospitalization	80 (39.6%)	-
Length of hospital stay	19.4 ± 17.7	-
≤10 days (n, %)	27 (34.2%)	-
>10 days (n, %)	52 (65.8%)	-

Continuous variables are shown as the median and interquartile range (IQR) (age, height, weight, post-COVID-19 time, and length of hospital stay). Abbreviations: BMI, body mass index.

**Table 2 biology-12-00749-t002:** Frequency of main symptoms estimated for the study groups.

Symptoms in Long COVID-19 (n, %)	N (%)		
	≤120 Days (n = 81)	>120 Days (n = 121)	*p*-Value
Fatigue	50.60%	59.50%	0.248
Myalgias	14.80%	24.80%	0.111
Anosmia/Hyposmia/Parosmia	4.90%	20.70%	0.002 *
Amnesia/recent memory loss/brain-fog	24.70%	13.20%	0.041 *
Shortness of breath	17.30%	20.70%	0.590
Headache	12.30%	15.70%	0.545
Arthralgias	3.70%	13.20%	0.026 *
Loss of hair	8.60%	13.20%	0.371
Chest pain	11.10%	14.90%	0.529
Muscle weakness	14.80%	11.60%	0.525
Decrease visual acuity	7.40%	5.00%	0.109
Paresthesia in limbs	7.40%	6.60%	1
Insomnia	3.70%	5.80%	0.743
Arrhythmias, bradycardias, tachycardias	6.20%	3.30%	0.488
Arterial hypertension	1.20%	4.10%	0.404
Hyperglycemia/Diabetes Mellitus	0.00%	4.10%	0.084
Anxiety	6.20%	6.60%	1
Irritation in the throat/throat	2.50%	4.10%	0.704
Dizziness	4.90%	3.30%	0.716
Itching and/or blemishes on the skin	3.70%	2.50%	0.685

Heatmap: highest frequency (red), lowest frequency (green) * Statistically significant if *p* < 0.05. Fisher Exact Test.

**Table 3 biology-12-00749-t003:** Lower limb temperatures and heart rate variability in study participants according to post-COVID-19 time.

Variables	Long COVID-19 Time (Days)			
	Control (n = 95)	≤120 Days (n = 81)	>120 Days (n = 121)	*p*-Value
**ROIs—Right foot**				
D1	29.70 (27.0–31.8)	29.70 (28.4–32.2) ^c^	28.80 (26.7–30.3)	0.001 *^#^
D2	27.60 (24.6–30.5)	28.70 (26.6–31.3) ^c^	27.20 (24.6–29.1)	0.003 *^#^
D3	27.70 (24.9–30.4)	28.80 (26.5–31.5) ^c^	27.30 (24.5–29.1)	0.003 *^#^
D4	27.70 (25.0–29.9)	28.60 (26.6–31.2) ^c^	27.00 (24.4–29.0)	0.003 *^#^
D5	29.40 (27.3–31.0) ^b^	29.80 (28.3–31.8) ^c^	28.50 (26.6–30.0)	<0.001 *^†^
ROIs—Left foot				
D1	29.40 (27.7–31.5) ^b^	30.00 (28.0–32.3) ^c^	28.50 (26.8–30.3)	<0.001 *^#^
D2	27.40 (25.2–30.6)	28.40 (26.3–31.3) ^c^	27.00 (24.8–29.8)	0.005 *^#^
D3	27.60 (25.4–30.9)	28.60 (26.7–31.5) ^c^	27.50 (24.9–29.9)	0.004 *^#^
D4	27.60 (25.6–30.4)	28.70 (26.7–31.7) ^c^	27.50 (24.8–29.6)	0.002 *^#^
D5	29.40 (28.1–31.2) ^b^	30.10 (28.3–31.9) ^c^	28.70 (26.9–30.1)	0.005 *^#^
HRV (Linear)				
iRR	852.0 (775–938) ^a^	810.0 (725–889)	842 (751–944)	0.038 *^†^
SDNN	38.5 (24.8–58.7) ^a,b^	25.0 (17.3–33.4)	25.4 (16.1–37.5)	<0.001 *^†^
RMSSD	44.8 (25.4–71.9) ^a,b^	26.5 (18.9–36.8)	24.4 (15.2–46)	<0.001 *^†^
LF	42.1 (28.3–61.5) ^a^	53.6 (42.5–66)	50.3 (36.7–61.5)	0.018 *^#^
HF	57.9 (38.5–71.7) ^a^	46.3 (34–57.4)	49.4 (38.5–63.2)	0.017 *^#^
LF/HF	0.73 (0.39–1.6) ^a^	1.16 (0.72–1.94)	1.02 (0.58–1.6)	0.018 *^†^
HRV (Non-Linear)				
SD1	32.6 (18.1–50.9) ^a,b^	18.7 (13.4–26)	17.3 (10.8–32.5)	<0.001 *^†^
SD2	44.8 (29.1–63.3) ^a,b^	29.5 (19.9–42.2)	28.5 (19.5–42.5)	<0.001 *^†^
SD1/SD2	0.69 (0.55–0.96) ^a^	0.57 (0.43–0.77)	0.63 (0.48–0.77)	0.004 *^†^
Entropy approximated	1.12 (0.16) ^a^	1.16 (1.09–1.21)	1.13 (0.14)	0.006 *^†^

Values are presented as medians with interquartile ranges (IQR). ROIs (regions of interest): D1 (distal end of the first toe to the navicular bone), D2, D3, and D4 (distal extremity of the second, third, and fourth toes up to the metatarsophalangeal joint), D5 (distal end of the fifth toe to cuboid bone); (iRR) mean RR interval, (SDNN) standard deviation of the normal RR interval (RMSSD) root mean square of the differences between adjacent normal RR intervals within a time interval, (LF) low frequency, (HF) high frequency, (LF/HF) low/high frequency ratio, SD1: Poincaré standard deviation perpendicular to the line of identity; SD2: Poincaré standard deviation along the line of identity. SD1/SD2: ratio of the short-to long-term range changes; entropy approximated the regularity of the RR interval series and signal complexity. * Statistically significant at *p* < 0.05. * with ^a^, ^b^, ^c^: ANOVA one criterion with Tukey’s post hoc test (parametric ^#^) or Kruskal–Wallis test with Dunn’s post hoc test (nonparametric ^†^). ^a^: control group vs. ≤120 days group (*p* < 0.05), ^b^: control group vs. >120 days group (*p* < 0.05), ^c^: ≤120 days group vs. >120 days group *: comparison between all groups, that is, control vs. long COVID.

## Data Availability

The data are not available due to privacy or ethical restrictions.

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
