# Peer review of "Imbalance of Peripheral Temperature, Sympathovagal, and Cytokine Profile in Long COVID"

_biology, 2023, doi:10.3390/biology12050749_

Round 1

Reviewer 1 Report

This study addresses physiological changes in long-COVID patients and evaluated molecular markers of inflammation in patients with long COVID. The authors assessed short-term heart rate variability (HRV), peripheral body temperature, and serum cytokine levels in patients with long COVID. The manuscript is well written but would benefit from more detailed descriptions of the figures, specifically figure 4.

Figure 1: the writing is hard to decipher and should be larger or the figure should be enlarged.

Line 229 and Fig. 3: just looking at the figure, it is hard to believe that significant differences exist for IL-17, IL-2 and IL-4, IL-6 given the large standard deviations.

Line 242 and Fig. 4: the figure needs more explanation both in the text and in the figure legend.  The figure legend should explain the two colors used and how to read the figure. Also, the axes are not labeled.

Author Response

Attached are the answers to the questions.

Reviewer 2 Report

This is a highly significant study on long-term COVID based on a brilliant notion of neurofunctional and immunological thermoregulation evaluation. It was meticulously carried out by a group of researchers with extensive academic expertise. Look out the comments highlighted below, especially in the discussion, where the authors might do a lot better.

METHOD

  1. It is critical to provide the technical characteristics of the thermal sensor employed in the text (thermal image resolution, frame rate, measurement accuracy, field of view, focus, minimum focal distance). The thermal camera used in the research is not medical equipment and is not calibrated for the evaluation of human beings. While it can be used to capture thermal images of human skin, it is not designed or validated for medical diagnosis, health assessment, or accurate body temperature measurement. The device is primarily intended for use in industrial applications, building inspection, leak detection and other similar applications. Therefore, it is important not to rely exclusively on the readings provided by this camera for body temperature assessment or any other medical purpose. In addition to being calibrated for temperature assessments with a temperature range of -20°C to 400°C, it produces a measurement accuracy of just 3°C (5.4°F) or 5% of the reading, which impacts the findings. I urge that writers aim to develop future studies in this area using more specialized sensors and software that are regulated by health organizations for usage in people. Although this does not diminish the study's findings or significance, it should be stated as a restriction in the discussion.
  2. The temperature color scale must accompany the thermal Figure 2. It is important that all infrared thermal images are accompanied by a temperature color scale so that the reader can properly interpret the information contained in the image. This is because the thermal image is made up of different colors that represent different temperature levels. Each color in the thermal image represents a temperature range, and the color scale provides a visual reference for these temperature ranges. The color scale is usually displayed along with the thermal image and is made up of a series of colors that vary from one extreme to the other. For example, the rainbow scale , this scale presents a color grading that starts with blue at lower temperatures, followed by green, yellow, orange, red and ending with white at higher temperatures. The rainbow of colors allows for easier interpretation of the thermal image, with temperature variations more evident and less prone to misinterpretation. This scale can also include numerical values that correspond to each color, which allows the reader to correlate colors with temperature values. Without the temperature color scale, the thermal image can be difficult to interpret correctly, as the colors can appear arbitrary and there would be no reference to get an idea of the actual temperature. An important detail is to keep the image's background monochromatic; typically, black is used to highlight only the temperature of the human skin. Therefore, the presence of the temperature color scale is critical to ensure a correct and accurate interpretation of the infrared thermal image.
  3. Fatigue is a common symptom in patients with COVID-19, but its definition may vary depending on the study and methodology used. Chalder Fatigue Scale , Bristol Fatigue Scale and the Multidimensional Fatigue Scale (MFI) can help measure fatigue in patients with COVID-19 and provide more objective and comparable data across different studies. What were the medical criteria used to define fatigue in the authors' study, as shown in Table 1?
  4. Visual changes in patients with long-standing COVID-19 can be assessed through ophthalmologic examinations, such as visual acuity, visual field, fundoscopy, and optical coherence tomography. Some studies have evaluated the presence of ocular symptoms, such as dry eye, conjunctivitis, photophobia and blurred vision, to assess the presence of visual changes in patients with long COVID. Some generic quality of life scales, such as the NEI-VFQ-25 questionnaire, may include questions about visual function and patient satisfaction with vision. What were the medical criteria used to define visual changes in this study, as shown in Table 1?
  5. There are several specific scales for assessing insomnia, including the Epworth Sleepiness Scale, the Pittsburgh Insomnia Scale (Pittsburgh Sleep Quality Index), the Sleep Quality Index (Sleep Quality Index), the Insomnia Severity Scale (Insomnia Severity Index) and the Stanford Sleepiness Scale Scale). In the long-term COVID literature, the Pittsburgh Sleep Scale Quality Index) seems to be one of the most used scales to assess insomnia. This scale consists of 19 items and assesses various aspects of sleep, including sleep quality, sleep latency, sleep duration, and use of sleep medication. What were the medical criteria used to define insomnia in this study, as shown in Tables 1 and 2?
  6. There are some neuropsychological scales that can be used to assess memory in general, such as the Wechsler Memory Scale and the Rivermead Memory Scale. What were the medical criteria used to define amnesia/loss of recent memory/lack of attention in this study, as shown in Table 2?
  7. Were clinical neurological evaluations performed on all participants (COVID and control group)? If so, please notify me because line 274 mentions neurological symptoms.
  8. Peripheral neuropathy is one of the neurological manifestations reported among 10.5% to 19% of patients with long-term COVID-19. There can be confusion between peripheral neuropathy and arthralgia, as both symptoms can cause pain in the extremities. It is important to carry out a careful clinical evaluation and, if necessary, additional tests to differentiate these conditions. Arthralgia is defined as joint pain without objective clinical signs of joint inflammation such as redness, swelling and warmth. There are some specific scales that can be used to assess arthralgia, such as the Visual Analogue Scale (VAS) and the Numerical Pain Scale (END). These scales allow the patient to assess the intensity of joint pain on a scale from 0 to 10. What were the medical criteria used to define arthralgia in this study, as shown in Table 2?
  9. Describe in detail how to replicate the authors' methodology for using temperature measurement lines on the thermal image of the feet. How did they get the selection over the image (anatomical details)? How were these temperature readings analyzed? Was the temperature calculated using the average, minimum, maximum, and deltas? To be more specific.
  10. Given that the authors are investigating an abnormal systemic COVID condition, what is the rationale for evaluating the fingers separately? Why compare the right and left feet? And why, even more, the back of the feet? The explanation may give the reader a better understanding of the technique's potential in the study of systemic diseases.

DISCUSSION

11.    Already inform in the first line of the discussion which alteration in the peripheral body temperature was found, do not leave it vacant. Inform values that are objective.

12.    In sentence 269, you could add that this is the first study to evaluate long-term COVID using thermography or something similar within the line of thought proposed in this sentence in relation to the literature.

13. Reinforce in the discussion the importance of how patients with long COVID were selected for the study, as the selection method can have an impact on the results.

  1. Reinforce in the discussion how the measurement of heart rate variability (HRV), peripheral body temperature and serum levels of cytokines was performed, as the method used may have an impact on the validity of the results.
  2. Explain to the reader in greater detail how the vagus system influences skin temperature, which is a sympathetic response. Yoga's physiological vagal stimulation raises skin temperature, for example, but the study is referring to a dysfunction of the vagal reflex system, sympathovagal balance, decreased parasympathetic modulation, instability, or even a decrease in skin temperature, as mentioned in Line 284? And why not in hemibody/hemisome (as suggested in line 305, implying thermal asymmetry in long COVID) or even body parts (feet, hands, face, and others)?
  3. There seems to be a misunderstanding in the text when discussing sympathetic hyperactivity, central effects, and peripheral effects, which should be revised. Sympathetic hyperactivity causes peripheral vasoconstriction in the skin despite an increase in central metabolism. Sympathetic overactivity is a state in which the sympathetic nervous system, which is responsible for regulating various bodily functions including heart rate (as measured in the study), blood pressure and body temperature, is overly active. This can lead to a number of effects on the body, both central and peripheral. The central effects of sympathetic overactivity include an increase in metabolic activity in various organs such as the heart, lungs and muscles. This can lead to increased heart rate, breathing and blood flow to these organs. However, the peripheral effects of sympathetic overactivity can include peripheral vasoconstriction in the skin, which can lead to decreased blood flow to the skin and a feeling of coldness or pallor. It is important to note that the effects of sympathetic overactivity can vary depending on the context and the specific conditions in which it occurs. Additionally, other factors such as body temperature regulation may also play a role in regulating blood flow to the skin and in responding to sympathetic overactivity. top of form
  4. The authors discuss the effect of cytokines on temperature in Line 280. What does this have to do with question 15 about the vagal reflex? Is it a different explanation or the same as what was discovered thermally?
  1. In light of the preceding question, why did the authors not conduct a thermal assessment of the entire body, given that this is a systemic disease or one involving the central nervous system? Looks like a limitation of the study.
  2. Why do the authors conclude that sympathetic modulation has increased (line 227)? Is it possible that you have peripheral sympathetic vasomotor hypotonia with vasodilation and elevated skin temperature? Make the distinction between what the authors actually discovered and what is postulated based on the literature and the authors' interpretation of the results clearer. Make the reader aware of these limits.
  3. What are the limitations of the study? The study was performed on a specific sample of patients with long-term COVID and may not be generalizable to other populations. In addition, the study is cross-sectional, which means that the results do not provide information about the evolution of the disease over time. Comment.
  4. In line 305, the authors discuss the evaluation of the thermal differential between the body's sides. To be logical, this sentence implies that there is asymmetry between the sides of the body in patients with long COVID, which may seem counterintuitive in a systemic inflammatory vascular disease. Is this asymmetry of 0.5°C discovered by the authors? Otherwise, despite the correct references, it would be better to revise this IRT comparison between the hemibodies (hemisomes) that would not fit in this discussion.
  5. What is the reasoning behind thermally modifying only the second, third, and fourth toes? Did you have either feet or a predisposition to laterality? Wouldn't the outcome be random (fluctuation, variation, noise)? Finding a statistically significant difference in a study does not necessarily mean that there is a biological explanation for that difference. Statistics is a tool for quantifying the difference between groups or variables and testing the probability that the observed difference is due to chance or other causes. The correct interpretation of statistical results must take into account other relevant information and knowledge. This may involve reviewing the scientific literature, considering the underlying biological hypotheses, and carefully evaluating the limitations and assumptions of the study. Statistics is a powerful tool in data analysis, but it is important to interpret results carefully and consider other relevant information to make accurate and reliable inferences, especially in expert-supported medical studies.
  6. I was puzzled when the authors stated that their findings are consistent with Gatt's study (60). Complement with objective data by numerically comparing study results with others.
  7. Could the study imply that there is a cutoff value for foot temperature that can be used to screen patients at higher risk of long-term COVID? Which temperature value and region were significantly stronger in inferring a cutoff value for screening or diagnosis of risk of having this pathological condition? Even if this was not the study's goal, it would be very interesting to leave this information in the text with certainty so that other researchers can continue to validate these very important findings. Even more important at this early stage than attempting to explain the actual cause of the temperature difference in the study.
  8. What are the clinical implications of the results? The authors should provide some treatment suggestions and prevention strategies based on the results, but it is important to assess whether these suggestions are supported by the study results.
  9. How does the study contribute to current knowledge about long COVID? Are the study results consistent with other studies on long COVID? Authors should further discuss how their thermal results compare with other studies in the literature.
  10. There is another very important aspect that was not addressed and could be added to this discussion, enriching the work and its citation in the literature. For example, IRT has been used to screen for fever in people with suspected COVID. However, it was a controversial method that had little impact on traceability because many infected people continued to circulate and infect others even when they did not have classic fever, which meant they went unnoticed by these monitoring IRT sensors (false negative). The authors' research focuses on the relationship between cytokines and temperature rise in patients with long-term COVID. And this is the most recent update on the subject. A study was recently published that demonstrated the importance of artificial intelligence software for analyzing thermal images in identifying people with possible COVID infection. The results showed that, despite peripheral vasodilation, these people did not have fever, but were identified by IRT evaluating all changes sympathetic and vagal vasomotor reactions related to the increase in cytokines, as well as the authors' well-documented findings. Reference: Infrared image method for possible COVID-19 detection through febrile and subfebrile people screening. J Therm Biol. 2023 Feb ;112:103444. doi : 10.1016/j.jtherbio.2022.103444.

CONCLUSION

28.    Indicate the magnitude of the temperature increase discovered by the authors in the study in the first line. Transmit quantitative data.

Some flaws that can be corrected include the need to better detail the methodologies used to measure the studied variables and the inclusion of a clearer and more detailed section on the limitations of the study. In addition, further discussion of the clinical implications of the results and a more detailed comparison with other studies in the literature may be useful to contextualize the results.

It was a pleasure to review this research. I apologize if my comments were too long and detailed, but I really liked your study and would like to see it published in the highest quality and rigor that your group is capable of.

Author Response

(The authors gave the same response as above.)
